# Nutritional Intervention in Patients with Multiple Sclerosis, Correlation with Quality of Life and Disability—A Prospective and Quasi-Experimental Study

**DOI:** 10.3390/neurosci6010004

**Published:** 2025-01-06

**Authors:** Konstantina Metaxouli, Chrysoula Tsiou, Eleni Dokoutsidou, Nikoletta Margari

**Affiliations:** Department of Nursing, University of West Attica, Egaleo, 12241 Athens, Greece; ctsiou@uniwa.gr (C.T.); edokout@uniwa.gr (E.D.); nmargari@uniwa.gr (N.M.)

**Keywords:** multiple sclerosis, functional disability, Mediterranean diet, quality of life

## Abstract

Multiple sclerosis (MS) is a multifactorial disease, with diet and lifestyle playing an important role in its development. The Mediterranean diet has been considered to be particularly beneficial for MS patients. The aim of the present study was to investigate the relationship between diet and MS, as well as evaluate the effect of the Mediterranean diet on patients’ quality of life and level of disability. The six-month study included 130 patients, divided into a control and intervention group. Data collection instruments were used for the collection of demographic and medical characteristics of the participants, as well as data regarding disability [(Multiple Sclerosis Rating Scale-Revised (MSRS-R) and Multiple Sclerosis Impact Scale-29 (MSIS-29)], nutrition [Mediterranean Diet Score (MedDiet Score) and Mini Nutritional Assessment (MNA)], and quality of life [Multiple Sclerosis Quality of Life-54 (MSQOL-54)]. The results indicated that the demographic characteristics of the groups were similar. The MNA score was positively associated with physical (*p* = 0.002) and mental health (*p* = 0.001). The intervention group reported an improvement in adherence to the Mediterranean diet, an increase in the MedDiet Score, and a decrease in the MSRS-R Score, indicating an improvement in functional capacity, nutritional status, and quality of life. In conclusion, the Mediterranean diet can improve the functionality and quality of life of patients with MS. Nutrition education is therefore deemed critical, and further research is required to reinforce these findings.

## 1. Introduction

Multiple sclerosis (MS) is a neurodegenerative, demyelinating, inflammatory, autoimmune disease of the central nervous system, whose incidence is increasing worldwide as a result of earlier diagnosis and improved patient management [1]. In the early stages of the disease, oxidative stress plays an important role in mitochondrial dysfunction, leading to damage to neurons and glia. Antioxidant agents are considered to decrease oxidative stress and perhaps protect against chronic demyelination and neuronal/axonal damage [2]. Thus far, it has been theorized that the incidence of the condition is directly correlated with specific geographic locations, as the prevalence of the disease seems to increase with the distance of a given region from the equator [3]. Other studies investigating population genetics have shown that the increased prevalence rates of the disease are not satisfactorily explained by mere genetics, suggesting that environmental risk factors play the most notable role in the development of multiple sclerosis [4]. Therefore, the causes of the disease appear to include both genetic and environmental factors [1]. Vitamin D deficiency, reduced sun exposure, stress, obesity, diet and infections are among the main environmental risk factors for MS [5].

There is considerable ambiguity with regard to dietary advice for people with multiple sclerosis, whereas the impact of dietary habits and lifestyle on the course of the disease remains obscure. Nutrition education is essential to provide knowledge and information about nutrition and healthy eating habits that are relevant to all people, whether they suffer from the disease or not. The aim of nutrition education is to enable people to make informed choices regarding their diet, understand the importance of different nutrients, and adopt healthier eating patterns to promote their overall well-being [6].

Nowadays, there is a considerable body of research that has investigated nutrition in relation to various diseases, such as heart conditions [7], diabetes, cancer [8], gastrointestinal disorders [9], obesity [10], and the nutrition of young people [11,12]. Although these topics have been extensively analyzed, the effect of diet on multiple sclerosis, especially in Greece, has been studied to a lesser extent [1].

The Mediterranean diet is a balanced diet that is usually accompanied by a healthier lifestyle [13]. It is an important and historical dietary pattern that has evolved in the Mediterranean basin, influenced by various cultures such as the Egyptians, the Greeks, the Romans, and others. Its basic components include olive oil, olives, whole grains, wine, vegetables, fruits, fish, and seafood, which have contributed to the amalgamation of a rich and healthy cuisine. Hippocrates recognized the importance of the diet for physical and mental health. The Mediterranean diet was enriched with new foods with the passage of time, as well as through interaction with various cultures. Today, it is considered a worldwide renowned diet associated with the culture, history, society, and lifestyle of the Mediterranean peoples. The consumption of olive oil is of central importance due to its beneficial properties, such as its neuroprotective action. Cereals, vegetables, legumes, and fruits are elements of the diet, as they are consumed on a daily basis, while meat consumption is moderate [14].

The present study aims to investigate and highlight the possible connection between the Mediterranean diet and multiple sclerosis, with the aid of a nutritional intervention. The research undertaken aims to improve our understanding of the interactions between diet and the condition, in order to identify possible associations and effects.

## 2. Materials and Methods

A study was conducted by enrolling participants in control and intervention groups at three points in time. The purpose of the study was to examine the dietary habits of patients with multiple sclerosis, whether these habits were related to their degree of disability, whether they affected their quality of life, how and to what extent disability impacted the patients’ quality of life, and whether nutritional education and adherence to the Mediterranean diet improved their quality of life. The participants were patients of the Neurological Outpatient Clinic and the Neurological Clinic of the General State Hospital of Nikaia, Piraeus. The sample consisted of 130 individuals selected via random sampling. Data collection was carried out from June 2021 to May 2022, with a random selection of patients based on the odd numbers on their appointment and admission cards. The intervention group (65 people) was given an educational sheet (pp. 14–15) on the Mediterranean diet and healthy practices for dealing with the disease. The diet regime included fish, fruits, vegetables, whole grains, and a restriction of red meat, dairy, and processed foods. After three and six months, the clinical status and quality of life of the participants were assessed. The study examined whether the condition of patients who adhered to the Mediterranean diet (as implied by the MedDiet score) improved or remained stable compared to three and six months before, and when compared to the control group. During the study, the intervention group remained unchanged, comprised of 65 patients, while the control group decreased from 72 to 65 people due to the withdrawal of six women and the random selection of one participant to equalize the groups. Inclusion criteria for the study were age ≥ 18 years, a good understanding of the Greek language, and signed informed consent. Patients receiving cortisone, patients who were recently diagnosed with multiple sclerosis, those who had participated in a pharmaceutical clinical trial, those who had recently changed their medication, or those who suffered from psychiatric diseases, intellectual disability, or comorbidities were excluded.

### 2.1. Questionnaires

For the purposes of the study, patients were fully informed and gave their written consent. They completed structured questionnaires through personal interviews lasting approximately half an hour in a quiet area of the hospital. Information was collected on social, demographic, and medical data, dietary habits, disability status, and quality of life. Data on gender, age, weight, height, marital status, education, employment status, financial status and place of residence were collected. In addition, data regarding years of illness, medication, cortisone administration, other health issues, exercise, smoking, and use of vitamins or nutritional supplements were obtained. Disability was assessed using the scales Multiple Sclerosis Rating Scale-Revised (MSRS-R) [15] and Multiple Sclerosis Impact Scale-29 (MSIS-29) [16]. Nutrition was assessed with the questionnaires Mediterranean Diet Score (MedDiet score) [17] and Mini Nutritional Assessment (MNA) [18]. For the assessment of quality of life, the Multiple Sclerosis Quality of Life-54 (MSQOL-54) was used [19]. The MSRS-R scale is a useful, brief tool for assessing the impact of multiple sclerosis on patients, providing a comprehensive understanding of the course of the disease and treatments. The MSIS-29 scale assesses the impact of the disease on daily life. The Mediterranean Diet Score questionnaire evaluates adherence to the Mediterranean diet and is a useful tool for assessing nutritional status. The Mini Nutritional Assessment (MNA) is a valid 18-question tool that assesses nutritional risk and nutritional status, suitable for use in geriatric assessment. The MSQOL-54 scale is utilized for the evaluation of quality of life of patients with multiple sclerosis, combining questions from the 36-Item health survey Short Form with 18 additional questions that are specific to multiple sclerosis.

### 2.2. Statistical Analysis

Using the Kolmogorov–Smirnov test, the distributions of the quantitative variables were tested for normality. Mean values and standard deviations (SD) were used for the description of quantitative variables. Absolute (n) and relative (%) frequencies were used to describe qualitative variables. To compare qualitative variables between the two groups, Pearson’s chi-square test or Fisher’s exact test was used. To compare quantitative variables between two groups, Student’s *t*-test for independent samples was utilized. To test the association between two quantitative variables, the Spearman correlation coefficient (rho) was used. Repeated-measures analysis of variance (ANOVA) was employed to test for differences in the under-study scales between the control and intervention groups and over time. Also, with the aforementioned method, it was assessed whether the degree of change over time of the under-study scales was different between the two groups. Repeated-measures analysis of variance (ANOVA) for the MNA and MSRS-R scales was performed using logarithmic transformations due to lack of normality. To check for type I error, due to multiple comparisons, the Bonferroni correction was used, whereby the significance level is 0.05/κ (κ = number of comparisons). Repeated-measures analysis of variance (ANOVA) was also utilized to evaluate the association between patients’ characteristics in the intervention group and the changes in all under-study scales. Significance levels are two-sided, and statistical significance was set at 0.05. The statistical program SPSS 26.0 was used for the analysis.

## 3. Results

The sample consisted of 130 patients with multiple sclerosis, divided into two equally sized (65 patients each) groups, i.e., control and intervention groups. The demographic and clinical characteristics for each group separately are presented in Table 1. The majority of patients in both groups were females, 81.5% for the control group and 67.7% for the intervention group. Additionally, 36.9% of patients in the control group were 41–50 years old, and 32.3% of patients in the intervention group were 31–40 years old. Most patients in both groups had a normal BMI (56.3% of the control group and 46.9% of the intervention group). Furthermore, 60% of the participants in the control group were married, and 50.8% of the intervention group were married. Sixty percent of the control group and 49.2% of the intervention group had children. Forty percent of the control group and 58.5% of the intervention group held a university degree. Moderate financial status was reported by 56.9% of the control group and 53.8% of the intervention group. The mean time since diagnosis was 11.7 years (SD = 7.7 years) for the control group and 9.7 years (SD = 6.6 years) for the intervention group. Under treatment were 90.8% of the control group and 96.9% of the intervention group. The characteristics of both groups were found to be similar (*p* > 0.05). 

Participants’ scores in MSRS-R, MNA, and MedDiet scales throughout the follow-up period, by group, are presented in Table 2. No significant differences regarding MSRS-R, MNA scores were found between the two groups at any time point (*p* > 0.05). No significant differences were found in MedDiet score between the two groups at baseline. However, at the 2nd (*p* = 0.007) and 3rd (*p* = 0.001) follow-ups, the intervention group had a significantly higher score, indicating that they were significantly more adherent to the Mediterranean diet pattern compared to the control group. Timewise, in the control group, no significant changes were found in MSRS-R, MNA, and MedDiet scales (*p* > 0.05). On the contrary, in the intervention group, there was a significant overall decrease (from time 1 to time 3) in MSRS-R score (*p* < 0.001), indicating a significant reduction in the functional difficulty experienced by the patients, as well as a significant overall increase in MNA total score (*p* = 0.023), indicating a significant improvement in patient nutrition, in the same time interval. MedDiet score in the intervention group increased significantly from one time point to the next, as well from time 1 to time 3 (*p* < 0.001), indicating a significant improvement in the dietary habits of the patients based on the Mediterranean pattern. The extent of change in MSRS-R (*p* = 0.006—Figure 1, p. 6, MNA (*p* = 0.044—Figure 2, p. 6), and MedDiet (*p* < 0.001—Figure 3, p. 7, scales differed significantly between the two groups. More specifically, no significant improvement was found in the control group, while in the intervention group, their functionality, feeding, and diet were improved significantly.

Participants’ scores in MSQOL-54 subscales throughout the follow-up period, by group, are presented in Table 3. Physical health, role limitations due to emotional problems, health perceptions, social function, sexual function, change in health, and satisfaction with sexual function presented no significant differences between the two groups at each timepoint (*p* > 0.05), had no significant time changes (*p* > 0.05), and no significant differences regarding the extent of change between the two groups (*p* > 0.05). Role limitations due to physical problems and health distress scores differed significantly between the two groups at time 1. Specifically, the intervention group had a significantly higher score in the role limitations due to physical problems subscale (*p* = 0.039) and a significantly lower score in the health distress subscale (*p* = 0.024). In the 2nd and 3rd measurements, the scores were similar for the two groups (*p* > 0.05). Timewise, no significant changes were found in role limitations due to physical problems and health distress scores in either group, and the degree of change was similar in both groups (*p* > 0.05). Pain score at time 1 was similar for the two groups. Then, at times 2 and 3, the pain score of patients in the intervention group was significantly higher compared to the control group (*p* = 0.035 and *p* = 0.022, respectively). Timewise, no significant changes were found in pain score, and the degree of change was similar in both groups. The emotional well-being score was similar between the two groups at each time point (*p* > 0.05). In the intervention group, the score remained at similar levels throughout the follow-up period, while in the control group, there was a significant decrease at time 3, indicating a deterioration of the patients’ quality of life in this domain, compared to time 1 (*p* = 0.010) and time 2 (*p* = 0.013). Consequently, the degree of change in emotional well-being score differed significantly between the two groups (*p* = 0.009). The energy score was similar between the two groups at each time point (*p* > 0.05). In the intervention group, the energy score remained at similar levels throughout the follow-up period, while in the control group, there was a significant decrease at time 3, indicating a worsening of the patients’ quality of life in this sector, compared to time 2 (*p* = 0.035). However, the extent of change in the energy score was similar in both groups (*p* > 0.05). Cognitive function and overall quality of life scores were similar between the two groups at each time point (*p* > 0.05). In the control group, cognitive function and overall quality of life scores remained at similar levels throughout the follow-up period, while in the intervention group there were significant increases at times 2 (*p* = 0.044 and *p* < 0.001, respectively) and 3 (*p* = 0.005 and *p* < 0.001, respectively), indicating an improvement in the patients’ quality of life in these specific domains, compared to time 1. As a consequence, the extent of change in cognitive function and overall quality of life scores differed significantly between the two groups (*p* = 0.049 and *p* = 0.006, respectively).

Participants’ composite scores of MSQOL-54 and MSIS-29 scales throughout the follow-up period, by group, are presented in Table 4. No significant differences were found between the two groups at each time point (*p* > 0.05). Participants’ scores in MSIS-29 scales did not change significantly throughout the follow-up period. However, in the mental health composite, the extent of change was found to be statistically significant (*p* = 0.050), since in the control group there was an increase and, in the intervention group, a decrease, yet not significant. The MSQOL-54 physical health composite score remained at similar levels throughout the follow-up period, while in the control group, there was a significant decrease at time 3, indicating a deterioration in the patients’ physical health, compared to time 2 (*p* = 0.012). The extent of change in the MSQOL-54 physical health composite score was similar in both groups (*p* > 0.05). The MSQOL-54 mental health composite score remained at similar levels throughout the follow-up, while in the control group, there was a significant decrease at time 3, indicating a worsening of the patients’ mental health, compared to time 1 (*p* = 0.050). The extent of change in the MSQOL-54 mental health composite score differed significantly between the two groups (*p* = 0.007), as presented in Figure 4 (p. 10).

### 3.1. Correlations Among the Changes in All Under-Study Scales in the Intervention Group

There was a significant positive correlation of the change in the MNA scale with changes in physical health (rho = 0.35; *p* = 0.004) and role limitations due to physical problems (rho = 0.33; *p* = 0.007). Thus, the more the patients’ nutrition improved, the more their quality of life improved in these domains. Moreover, the more the patients’ physical health improved (based on the MSIS-29 scale), the more their functional difficulty decreased (rho = 0.27; *p* = 0.030) and the more their nutrition improved (rho = −0.25; *p* = 0.045).

### 3.2. Association of the Changes in All Under-Study Scales in the Intervention Group with Their Characteristics

It was found that the change in MedDiet score presented a statistically significant difference between the two genders (*p* = 0.047), as men (Mean = 1.6; SD = 4.3) had a significantly smaller increase when compared to women (Mean = 3.6; SD = 3.6). Moreover, it was found that the change in the overall quality of life subscale of MSQOL-54 differed significantly between patients with children and patients without (*p* = 0.042), as the former (Mean = 1.1; SD = 3.5) reported a significantly smaller increase compared to the latter (Mean = 3.6; SD = 5.0). No other significant associations were found between the changes in under-study scales and patients’ characteristics in the intervention group.

## 4. Discussion

### 4.1. Participation and Group Characteristics

Participation in all three follow-ups was excellent in both the intervention and control groups, with the characteristics of the two groups being similar (*p* > 0.05). Multiple sclerosis, a chronic inflammatory disease of the central nervous system, appears to affect women and Caucasian people disproportionately [20]. In the present study, women comprised 81.5% of the control group and 67.7% of the intervention group (Table 1, p. 4 ). Voskuhl et al. conducted a study with mice with the aim to examine the effect of gender on neurodegeneration and autoimmunity. Their findings concluded that T lymphocytes may play a role. Women, due to hormonal modifications, are considered susceptible to multiple sclerosis, but men may be at risk for a faster and more severe progression of the disease [21]. In the postmenopausal period, the inflammatory activity of the disease appears to decrease, while it is argued that the female peripheral immune system responds more effectively, limiting the activity of the disease [22,23].

### 4.2. Education

In the present study, university graduates constituted 40.0% of the control group and 58.5% of the intervention group (Table 1, p. 4). A higher educational level is associated with positive disease progression and improved quality of life indicators. Patients with higher academic achievements seem to have better memory and information processing abilities, as well as well-being, something that potentially facilitates patients with multiple sclerosis in seeking better job opportunities [24].

### 4.3. Income

In the present study, 56.9% of the control group and 53.8% of the intervention group had a moderate income (Table 1, p. 4). A study by Vozikis and Sotiropoulou in Greece showed that patients face financial uncertainty due to high healthcare costs, with costs ranging from EUR 3629 to 22,800 per year [25]. The overall financial burden of multiple sclerosis is high, affecting the quality of life of patients. Ellenberger et al. presented a study in which differences were found among the countries investigated (Germany, Poland, Sweden, and the UK). This finding underlines the lack of equality at the European level, which in turn seems to affect the progression of patients’ disability [26].

### 4.4. Pain

The total scores of pain, which is one of the most common symptoms of the disease, did not present significant changes, with the two groups experiencing similar levels of pain. According to a study by Junqueira et al., pain negatively affected the quality of life of patients, presenting a negative relation to their physical activity and mood [27]. Exercise and a healthy diet seem to reduce pain levels among MS patients. As reported by Strober et al., a balanced diet that provides adequate nutrients can positively affect the patients’ well-being [28].

### 4.5. Emotional Well-Being

The emotional well-being score in the third measurement presented a statistically significant decrease, showing a deterioration in the patients’ quality of life. The rate of change in the emotional well-being score was different between the two groups (*p* = 0.009, Table 2, p. 5). A six-month study that evaluated the effectiveness of a wellness intervention, which also included a dietetic intervention (paleolithic, vegetarian, Mediterranean), found that the intervention group showed a statistically significant improvement in quality of life and a reduction in fatigue [29].

### 4.6. Cognitive Function and Overall Quality of Life

In the domains of cognitive function and overall quality of life, scores in the intervention group increased (2nd measurement *p* = 0.044 for cognitive function and *p* < 0.001 for overall quality of life; 3rd measurement *p* = 0.005 for cognitive function and *p* < 0.001 for overall quality of life, Table 2, p. 5). Cognitive difficulties and depression seem to concern a significant number of patients and are a common feature that affects the quality of life of patients with a study reporting that the prevalence of cognitive impairment ranges from 20% to 88% depending on the type of multiple sclerosis [30]. Research related to the cognitive decline of patients showed that cognitive function is affected by a high body mass index (BMI), which negatively affects patients’ quality of life [31].

### 4.7. Mental Health

The patients in the control group showed a deterioration in mental health. The extent of change in the total mental health score on the MSQOL-54 (Table 3, p. 8) differed between the two groups (*p* = 0.007). A study with 6989 participants linked diet to disability and severity of symptoms in patients with multiple sclerosis. Patients with improved diet quality showed lower levels of disability as well as depression, while a healthy lifestyle reduced fatigue and depression [32].

### 4.8. Significance of Nutrition in Multiple Sclerosis

The MedDiet scale score in the intervention group (*p* = 0.007 and *p* = 0.001, Table 4, p. 10) revealed higher rates of adherence to the Mediterranean diet. Additionally, in the intervention group, the MSRS-R score (*p* < 0.001, Table 4, p. 10) showed a decrease in functional impairment, the MNA score (*p* = 0.023, Table 4, p. 10) showed an improvement in nutrition, and the MedDiet score (*p* < 0.001, Table 4, p. 10) showed an improvement in the patients’ dietary habits based on the Mediterranean pattern. Furthermore, in the intervention group, as the patients’ nutrition improved, their quality of life improved, and conversely, as the patients’ physical health improved (MSIS-29 scale), their functional impairment decreased (rho = 0.27; *p* = 0.030). The control group showed a decrease in the physical health score (MSQOL-54) at the 3rd measurement, indicating a deterioration of the patients’ physical health compared to the 2nd measurement (*p* = 0.012). A study by Yadav et al. highlighted the importance of nutrition in the development and improvement of multiple sclerosis [33]. A healthy and balanced diet was associated with milder symptoms and disease progression as reported by Simpson-Yap et al. [34]. Individuals who did not follow a specialized diet program seemed to be more likely to be obese and more prone to developing progressive multiple sclerosis [35]. Cantoni et al. reported that, in the context of multiple sclerosis, dietary restriction may contribute positively to the disease due to its neuroprotective effect, where specific dietary interventions for the prevention and treatment of the disease are mentioned [36]. Dietary factors have been shown to influence the incidence, severity of symptoms and progression of multiple sclerosis. The role of specific dietary factors, such as higher intake of processed meat, was associated with an increased risk of multiple sclerosis, as presented in a study by Ghazavi et al. in Iran [37]. For patients with multiple sclerosis, it is particularly important to follow a balanced diet, avoiding a high BMI that may cause problems in the progression of the disease [38]. Incorporating the Mediterranean diet into patients’ daily life can be a particularly effective strategy in terms of improving the quality of life and overall health of people suffering from multiple sclerosis. Besides, as demonstrated by the images–diagrams on pages 6, 7 and 10, and, patients who follow this dietary pattern show significant improvements in their well-being and physical condition.

As far as the finding concerning the intervention group and its compliance with the Mediterranean diet is concerned, it appears that men showed a smaller increase in MedDiet score compared to women, something that is not supported by the evidence in the extant literature. This may be explained by different dietary preferences, and more specifically by the fact that men may have a greater preference for meat or high-protein foods, which are not included among the staples of the Mediterranean diet. Another explanation could be the decreased flexibility of men’s dietary habits when compared to women, and therefore their difficulty in adapting to new dietary practices such as those associated with the Mediterranean diet. A further explanation may be the level of interest and discipline, with women possibly being more disciplined in adopting healthy eating habits compared to men. Finally, another explanation could be the influence of social factors, such as peer pressure or social expectations, which may influence the eating behaviors of men and women differently.

Furthermore, it was found that the change in the overall quality of life subscale of the MSQOL-54 differed significantly between patients with and without children. Nevertheless, this finding, which was not supported by evidence in the relevant literature, could be interpreted. One reason could be the time availability of patients without children, who have more free time to focus on monitoring their diet and implementing the proposed changes. Another possible explanation is that these patients may have more resources to buy healthy foods or follow nutritional advice. Undoubtedly, when there is a supportive environment, the provided encouragement and support ensure improved compliance with healthy eating habits.

## 5. Conclusions

The Mediterranean diet appears to have a positive effect on the quality of life, cognitive function and functional capacity of patients with multiple sclerosis. The Mediterranean diet intervention was associated with an improvement in patients’, while the control group showed a decrease in their quality of life. The MNA score was positively associated with physical and mental health. Furthermore, it was found that the change in the MedDiet nutritional score differs significantly between men and women, with women presenting a greater improvement. Finally, the perceived quality of life in patients without children was higher compared to those with children. The adoption of the Mediterranean diet may be an effective approach to improving the quality of life and general health of patients with multiple sclerosis.

## 6. Research Limitations

The findings of the present study are limited by the small sample size and the single healthcare setting. The specific hospital was chosen because it had a well-organized Neurology Clinic and Outpatient Clinics that monitor a significant number of multiple sclerosis patients. This choice allowed researchers to focus on their objectives efficiently, as the hospital already had an established system for treating and monitoring such patients. The pandemic raised obstacles for data collection, as many patients avoided in-person hospital visits and used remote prescription services instead. This further reduced the potential sample size for the study. Despite the challenges, the participants who were included showed a positive attitude toward the study and were willing to participate, reflecting strong engagement. Due to the small sample size and the study being confined to one hospital, the findings cannot be reliably generalized to all multiple sclerosis patients in Greece. The results are considered indicative, meaning they provide useful insights but are not definitive or representative of the entire MS patients population. Furthermore, the self-assessment of individuals with multiple sclerosis, on all scales, was subjective and therefore may have introduced systematic bias. Furthermore, the association between diet and functional disability could have been demonstrated in a more straightforward manner if the patients’ anti-inflammatory or antioxidant parameters were measured, an analysis that was practically impossible in the context of this study. Also, no biochemical parameters were taken into consideration that might have seemed useful for portraying the general condition of the participants.

## 7. Mediterranean Diet Information Sheet 

Diet is a way of life and should be combined with water consumption, exercise, and good sleep hygiene. Enjoyment and variety in diet are essential. But what is deemed more essential is the maintenance of a healthy body weight (Figure 5).

Olive oil is the cornerstone of the Mediterranean diet, which is why it is advised to be used in cooking.

### 7.1. Additionally, You Should

Always have breakfast.Avoid eating out frequently.Avoid processed meals and snacks.Avoid increased salt addition.Avoid excessive alcohol (a glass of wine is considered good) and soft drinks.Consume lots of fresh, seasonal fruits and vegetables.Consume fewer legumes and more oily fish.Consume unsalted nuts.Consume wholegrain cereals/pasta, potatoes, brown rice, multigrain bread.Consume fish (moderate to high consumption).Consume dairy products in moderation.Low consumption of red meat.Low consumption of sweets and sugar [39,40].

### 7.2. Advices for Patients with Multiple Sclerosis

BESIDES HEALTHY DIET, YOU SHOULD ALSO PAY ATTENTION TO:Stress is not advisable.Exercise is essential.Relaxation is essential.A good night’s sleep is essential.Avoid smoking, alcohol.Avoid extreme temperatures.Sun exposure is prohibited during the summer months.Adherence to medication.Go through medical appointments–examinations.It is essential to keep a diary, where you will record anything related to the medical record—medical visits and hospitalizations.Adopt an optimistic way of thinking [41].

## Figures and Tables

**Figure 1 neurosci-06-00004-f001:**
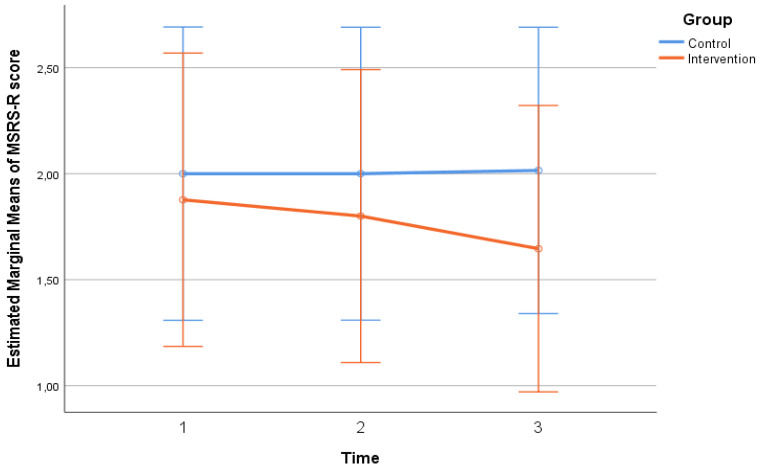
The changes in MSRS-R scale, three times between the two groups.

**Figure 2 neurosci-06-00004-f002:**
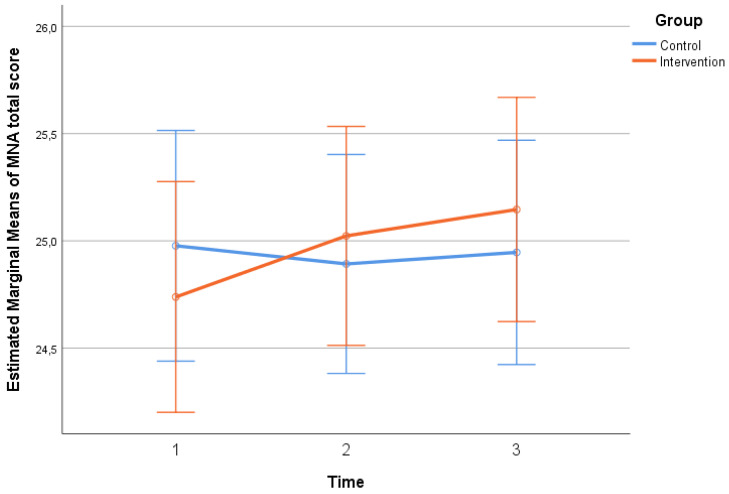
The changes in MNA total score scale, three times between the two groups.

**Figure 3 neurosci-06-00004-f003:**
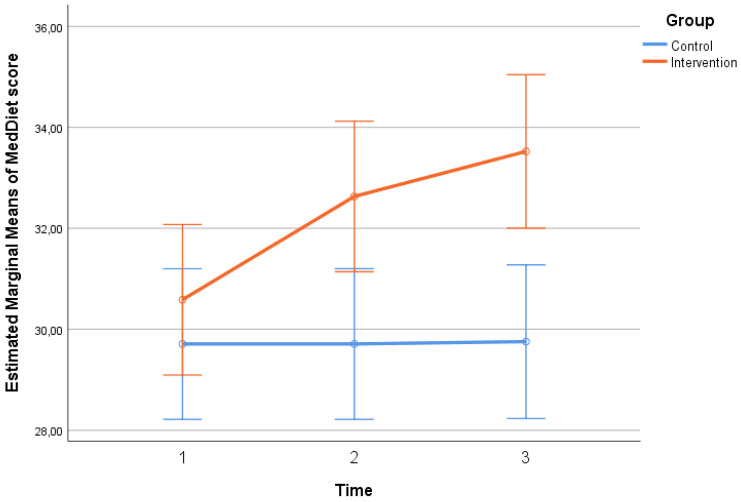
The changes in MedDiet score scale, three times between the two groups.

**Figure 4 neurosci-06-00004-f004:**
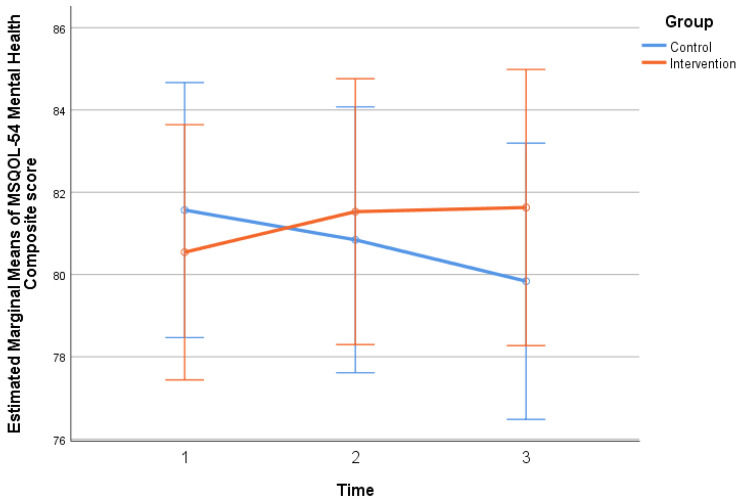
The changes in composite score in mental health of MSQOL-54 scale, three times between the two groups.

**Figure 5 neurosci-06-00004-f005:**
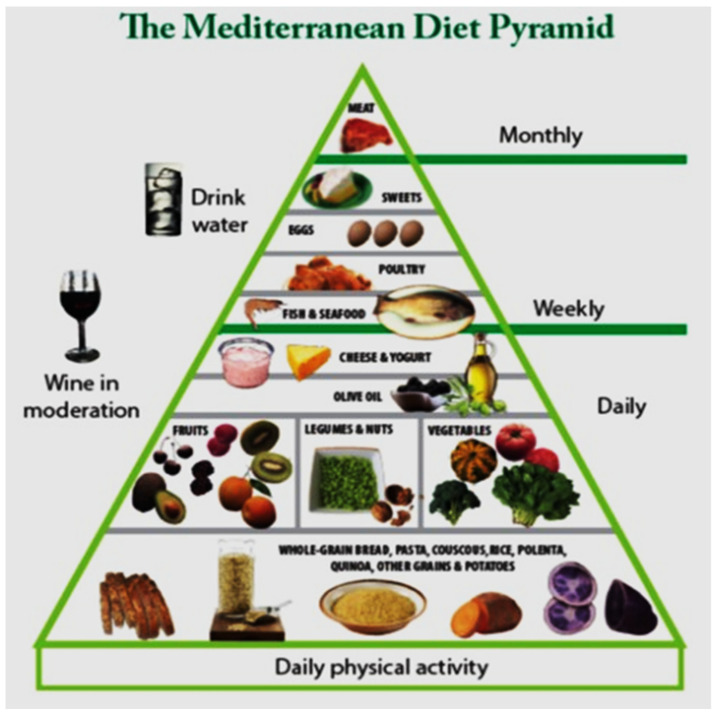
Mediterranean diet pyramid (Available from: web@agcenter.lsu.edu, 24 December 2024).

**Table 1 neurosci-06-00004-t001:** Sample characteristics, by group.

		Group		*p*
		Control (*n* = 65; 50%)	Intervention (*n* = 65; 50%)	
		n (%)	n (%)	
Gender				
	Women	53 (81.5)	44 (67.7)	0.070 +
	Men	12 (18.5)	21 (32.3)	
Age				
	18–30	6 (9.2)	7 (10.8)	0.771 ++
	31–40	19 (29.2)	21 (32.3)	
	41–50	24 (36.9)	17 (26.2)	
	51–60	15 (23.1)	19 (29.2)	
	61+	1 (1.5)	1 (1.5)	
BMI, mean (SD)		25.6 (5.1)	25.8 (4.6)	0.876 ‡
BMI				
	Normal	36 (56.3)	30 (46.9)	0.419 +
	Overweight	18 (28.1)	25 (39.1)	
	Obese	10 (15.6)	9 (14.1)	
Family status				
	Unmarried	21 (32.3)	25 (38.5)	0.769 ++
	Married	39 (60)	33 (50.8)	
	Widowed	1 (1.5)	2 (3.1)	
	Divorced	4 (6.2)	5 (7.7)	
Children		39 (60.0)	32 (49.2)	0.218 +
Educational level				
	Primary school	1 (1.5)	1 (1.5)	0.090 ++
	Middle school	6 (9.2)	3 (4.6)	
	High school	29 (44.6)	17 (26.2)	
	University	26 (40.0)	38 (58.5)	
	MSc/PhD	3 (4.6)	6 (9.2)	
Working status				
	Unemployed	12 (18.5)	15 (23.1)	0.691 ++
	Employed	35 (53.8)	35 (53.8)	
	Pensioner	13 (20)	13 (20)	
	Household	5 (7.7)	2 (3.1)	
Financial status				
	Poor	4 (6.2)	7 (10.8)	0.639 +
	Moderate	37 (56.9)	35 (53.8)	
	Good	24 (36.9)	23 (35.4)	
	Very good	0 (0.0)	0 (0.0)	
	Excellent	0 (0.0)	0 (0.0)	
Attend physical exercise program		28 (43.1)	25 (38.5)	0.592 +
Smoking		26 (40)	32 (49.2)	0.290 +
Vitamins		33 (50.8)	32 (49.2)	0.861 +
Years with disease, mean (SD)		11.7 (7.7)	9.7 (6.6)	0.105 ‡
Under treatment		59 (90.8)	63 (96.9)	0.273 ++

+ Pearson’s chi-square test; ++ Fisher’s exact test; ‡ Student’s *t*-test.

**Table 2 neurosci-06-00004-t002:** Participants’ scores in MSRS-R, MNA, and MedDiet scales throughout the follow-up period, by group.

	Group	Time 1	Time 2	Time 3	Change from Time 1 to 3	*p* ^2^			*p* ^3^
		Mean (SD)	Mean (SD)	Mean (SD)	Mean (SD)	1 vs. 2	2 vs. 3	1 vs. 3	
MSRS-R score	Control	2.00 (3.09)	2.00 (3.08)	2.02 (3.02)	0.02 (0.54)	>0.999	>0.999	>0.999	0.006
	Intervention	1.88 (2.52)	1.80 (2.52)	1.65 (2.45)	−0.23 (0.49)	0.067	0.109	<0.001	
	*p* ^1^	0.635	0.896	0.759					
MNA total score	Control	25.0 (2.1)	24.9 (2.1)	24.9 (2.2)	0.0 (1.1)	>0.999	>0.999	>0.999	0.044
	Intervention	24.7 (2.3)	25.0 (2)	25.1 (2.1)	0.4 (1.3)	0.060	>0.999	0.023	
	*p* ^1^	0.513	0.713	0.583					
MedDiet score	Control	29.7 (6.0)	29.7 (5.5)	29.8 (5.9)	0.0 (3.1)	>0.999	>0.999	>0.999	<0.001
	Intervention	30.6 (6.1)	32.6 (6.6)	33.5 (6.5)	2.9 (3.9)	<0.001	0.004	<0.001	
	*p* ^1^	0.412	0.007	0.001					

Note: Analysis for MSRS-R score and MNA total score was based on logarithmic transformations. ^1^ *p*-value for group effect; ^2^ *p*-value for time effect; ^3^ repeated measures ANOVA *p*-value regarding timegroup effect.

**Table 3 neurosci-06-00004-t003:** Participants’ scores in MSQOL-54 subscales throughout the follow-up period, by group.

	Group	Time 1	Time 2	Time 3	Change from Time 1 to 3	*p* ^2^			*p* ^3^
		Mean (SD)	Mean (SD)	Mean (SD)	Mean (SD)	1 vs. 2	2 vs. 3	1 vs. 3	
Physical health	Control	81.3 (23.6)	83.6 (20.9)	83.5 (21.3)	2.2 (13.3)	0.154	>0.999	0.194	0.315
	Intervention	83.4 (21.5)	83.9 (21.7)	83.9 (21.8)	0.5 (3.2)	>0.999	>0.999	>0.999	
	*p* ^1^	0.601	0.934	0.919					
Role limitations due to physical problems	Control	81.2 (29.3)	81.2 (31.6)	80.4 (32.6)	−0.8 (10.8)	>0.999	0.309	>0.999	0.512
	Intervention	91.2 (25.2)	89.6 (26.1)	90 (26.1)	−1.2 (11.2)	0.755	>0.999	>0.999	
	*p* ^1^	0.039	0.098	0.066					
Role limitations due to emotional problems	Control	87.7 (25.4)	85.6 (28.2)	84.6 (30.1)	−3.1 (16.4)	0.395	0.846	0.194	0.379
	Intervention	92.8 (22.4)	91.8 (23.6)	92.3 (23.4)	−0.5 (9.3)	>0.999	>0.999	>0.999	
	*p* ^1^	0.225	0.180	0.106					
Pain	Control	81.7 (25.3)	81.6 (25.5)	81 (25.7)	−0.7 (5.6)	>0.999	0.595	0.777	0.236
	Intervention	88.0 (17.4)	89.7 (17.5)	89.8 (16.9)	1.8 (4.2)	0.579	0.550	>0.999	
	*p* ^1^	0.100	0.035	0.022					
Emotional well-being	Control	78.8 (16.9)	77.7 (17.4)	75.4 (17.8)	−3.4 (9.2)	0.535	0.013	0.010	0.009
	Intervention	74.9 (14.0)	76.2 (11.4)	75.8 (10.3)	0.9 (9.5)	0.359	>0.999	>0.999	
	*p* ^1^	0.150	0.585	0.866					
Energy	Control	71.9 (18.0)	72.3 (17.8)	70.3 (18.7)	−1.7 (10.7)	>0.999	0.035	0.484	0.329
	Intervention	72.9 (11.4)	72.4 (10.9)	72.4 (9.8)	−0.5 (8.1)	>0.999	>0.999	>0.999	
	*p* ^1^	0.710	0.962	0.413					
Health perceptions	Control	36.0 (16.6)	35.7 (17.3)	35 (17.6)	−1.0 (4.5)	>0.999	0.223	0.067	0.406
	Intervention	31.2 (15.5)	29.8 (15.0)	29.8 (14.9)	−1.5 (5.8)	0.066	>0.999	0.073	
	*p* ^1^	0.083	0.061	0.070					
Social function	Control	86.4 (19.0)	86.2 (20.3)	85.9 (21)	−0.5 (6.9)	>0.999	>0.999	>0.999	0.458
	Intervention	89.6 (19.3)	89.9 (18.8)	90.1 (18.1)	0.5 (6.4)	>0.999	>0.999	>0.999	
	*p* ^1^	0.342	0.280	0.221					
Cognitive function	Control	93.2 (14.1)	94.3 (13.4)	93.8 (13.5)	0.7 (5.2)	0.125	>0.999	>0.999	0.049
	Intervention	93.6 (13.5)	95.0 (11.1)	96.4 (8.8)	2.8 (8.2)	0.044	0.062	0.005	
	*p* ^1^	0.849	0.749	0.206					
Health distress	Control	76.6 (18.2)	75.9 (18.2)	75.2 (19.1)	−1.5 (7.8)	>0.999	0.648	0.503	0.122
	Intervention	69.0 (19.9)	70.8 (17.3)	69.7 (16.8)	0.7 (9.1)	0.203	0.253	>0.999	
	*p* ^1^	0.024	0.101	0.086					
Sexual function	Control	77.4 (33.5)	78.6 (32.1)	78.0 (32.7)	0.6 (6.5)	>0.999	0.788	>0.999	0.193
	Intervention	76.2 (31.8)	77.4 (31.6)	77.4 (30.7)	1.2 (7.8)	0.224	>0.999	0.568	
	*p* ^1^	0.667	0.823	0.914					
Change in health	Control	46.9 (13.6)	46.9 (13.6)	46.9 (13.6)	-	>0.999	>0.999	>0.999	0.617
	Intervention	48.5 (13.2)	48.5 (13.2)	49.2 (12.5)	0.8 (8.8)	>0.999	0.755	0.963	
	*p* ^1^	0.514	0.514	0.316					
Satisfaction with sexual function	Control	72.3 (32.2)	71.5 (33.3)	71.5 (33.3)	−0.8 (6.2)	0.851	>0.999	>0.999	0.084
	Intervention	62.7 (34.8)	63.8 (34.2)	63.8 (34.8)	1.2 (6.9)	0.327	>0.999	0.475	
	*p* ^1^	0.105	0.197	0.200					
Overall quality of life	Control	72.0 (13.9)	72.2 (14.2)	72.6 (15.2)	0.6 (4.2)	>0.999	0.500	0.773	0.006
	Intervention	71.4 (11.8)	73.5 (11.2)	73.7 (11.3)	2.4 (4.5)	<0.001	>0.999	<0.001	
	*p* ^1^	0.779	0.559	0.631					

^1^ *p*-value for group effect; ^2^ *p*-value for time effect; ^3^ repeated measures ANOVA *p*-value regarding timegroup effect.

**Table 4 neurosci-06-00004-t004:** Participants’ composite scores of MSQOL-54 and MSIS-29 scales throughout the follow-up period, by group.

	Group	Time 1	Time 2	Time 3	Change from Time 1 to 3	*p* ^2^			*p* ^3^
		Mean (SD)	Mean (SD)	Mean (SD)	Mean (SD)	1 vs. 2	2 vs. 3	1 vs. 3	
MSQOL-54									
Physical Health Composite Score	Control	72.3 (17.5)	73.4 (16.9)	72.7 (17.8)	0.4 (4.5)	>0.999	0.012	0.869	0.362
	Intervention	73.4 (14.8)	73.4 (14.9)	73.4 (14.6)	0.0 (2.6)	>0.999	>0.999	>0.999	
	*p* ^1^	0.964	0.979	0.830					
Mental Health Composite Score	Control	81.6 (13.8)	80.8 (14.7)	79.8 (15.8)	−1.7 (7.1)	0.487	0.082	0.050	0.007
	Intervention	80.5 (11.4)	81.5 (11.5)	81.6 (11.1)	1.1 (4.4)	0.173	>0.999	0.425	
	*p* ^1^	0.645	0.767	0.457					
MSIS-29									
Physical Health Composite Score	Control	7.67 (13.38)	7.73 (13.33)	7.67 (13.32)	0.00 (1.95)	>0.999	>0.999	>0.999	0.112
	Intervention	6.19 (12.92)	5.6 (12.05)	5.58 (12.09)	−0.61 (2.63)	0.077	>0.999	0.103	
	*p* ^1^	0.522	0.340	0.350					
Mental Health Composite Score	Control	10.17 (15.23)	10.64 (15.51)	11.24 (16.64)	1.07 (5.66)	>0.999	0.362	0.458	0.050
	Intervention	13.68 (14.32)	12.91 (13.02)	12.56 (13.08)	−1.11 (6.30)	0.606	>0.999	0.412	
	*p* ^1^	0.179	0.369	0.615					

^1^ *p*-value for group effect; ^2^ *p*-value for time effect; ^3^ repeated measures ANOVA *p*-value regarding timegroup effect.

## Data Availability

The data presented in this study are available on request from the corresponding author due to privacy reasons.

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
