# Peer review of "Nutritional Intervention in Patients with Multiple Sclerosis, Correlation with Quality of Life and Disability—A Prospective and Quasi-Experimental Study"

_neurosci, 2025, doi:10.3390/neurosci6010004_

Round 1
Reviewer 1 Report
Comments and Suggestions for Authors
The article is interesting. However, some important drawbacks should be revised.
In the introduction, more specific information on the neuropathology of MS should be given and on the factors regulating myelin regeneration.
Considering the credibility of the evidence for all direct comparisons is very low because most of the trials have a high or moderate risk of bias and small sample sizes are not significant, as sustained in a meta-analysis (Linda G Snetselaar et al. Neurology, 2023.), it isn't easy to draw some conclusions from this study because some informations about the diet (calories and composition) are lacking. Not only the quality of diet (MedDiet), but also the quantity of food is necessary to understand the biochemical processes ( Bianchi VE. Nutr Neurosci. 2021). Certain foods may decrease the risk or influence the progression of MS, such as increased intake of fish or polyunsaturated fatty acids, caloric restriction, and fasting-mimicking diets, while saturated fat ingestion may increase the progression (Monica R Langley et 2020).
It is important to know the carbohydrate intake, fat, and polyphenols ingestion.
Unfortunately, no information on biochemical parameters is given between the patient's data. Insulin resistance? Cholesterol? Triglycerides?
Do the patients change their body weight? or body composition?
The evaluation of physical exercise is utmost important.
Furthermore, the article is too long and should be reduced.
Line 53: "Nowadays, there is a considerable body of research that has investigated nutrition in relation to various diseases, such as heart conditions, diabetes, cancer, gastrointestinal dis-.." reference is lacking.
Line 58: "The Mediterranean Diet is a balanced diet that is usually accompanied by a healthier.." explain the criteria of MedDiet evaluation.
Line 62: explain the " rich and "healthy cuisine...." what is.
Line 91: how did you control patients' adherence to the Mediterranean diet? Which questionnaire? (Martínez-González MA, et al. Plos one. 2012). The Mini Nutritional Assessment has been associated with a high-risk ‘overdiagnosis and is highly sensitive in identifying patients at risk of malnutrition and low cost-benefit (Cereda, E. Curr Opin Clin Nutr Metab Care. 2012). In this particular neurological disease is not the best choice because the quantity of food is not expressed.
Nowadays, how dietary factors modulate oligodendrocyte biology, myelin injury, and myelin regeneration remains unexplained, and the authors should try to give some suggestions.
The influence on the reduction in inflammatory conditions and the amount of reactive oxygen species (ROS), the restoration of the myelin sheath of the neurons, and the regeneration of mitochondria. Insulin activity is essential (also sex hormones and IGF-1 play a crucial role). Is the ketogenic diet effective? (Damian Dyńka et al. Nutrients. 2022) (Ogata T, et al. Mol Neurobiol. 2006)
In the conclusion, although some important benefit in the quality of live of this group of MS patients, no clear evidence of MedDiet on biochemical and physical parameters is given.
Author Response
Dear reviewer,
I would like to thank you for your prompt response, as well as for your insightful comments.

Reviewer 2 Report
Comments and Suggestions for Authors
I believe that this is an interesting analysis of Mediterranean diet and the prevalence of Multiple Sclerosis.
However, some issues that may need to be addressed and to be taken into account are listed below.
1. “The research questions developed in the context of the study concerned the dietary habits of patients with multiple sclerosis, whether they were associated with the degree of their disability, whether they affected their quality of life and potential disability and to what extent, as well as whether nutritional education and adherence to the Mediterranean diet improved their quality of life and their disability status.” I believe that the research questions that have been tried to be covered by this study are too extensive and I believe that neither the sample and the duration of the study of the population can cover all these aspects. Thus, I propose their minimization, especially in case that there have not been deducted concluding and direct results about them.
2. The authors state “The diet regime included fish, fruits, vegetables, whole grains, and a restriction of red meat, dairy and processed foods.” How did you monitor the patients assuring their compliance?
3. If this was their diet, why is it stated as Mediterranean? It was stated arbitrarily or based on some rules? If there were certain rules these should be stated, or the type of diet should be referred in a different manner.
4. In the same manner the title refers to nutritional interventions in general. If it is a Mediterranean diet it should be stated also.
5. For the “Mediterranean diet information sheet” and table and advices at the end of the text, I do not understand in some cases its value. Furthermore, most of the dictated in these tables proposals of life should be followed in the article by relevant references that led to the production of this certain sheet with these advises. Thus, additional reference to bibliography and more extensive analysis should be required for the addition and approval of such sheet.
6. In the text “As far as the finding concerning….. men and women differently” and in other relevant cases some references should be added supporting the authors claims.
7. In the discussion section I can not decipher the reason of reference to the education, income and of the gender, if the findings do not correlate with these factors.
Author Response
Dear Reviewer,
I would like to thank you for your prompt response, as well as your insightful comments.

Round 2
Reviewer 1 Report
Comments and Suggestions for Authors
The manuscript has been improved, after revision, however the lacking of any metabolic information or body weight changes is a limiting scientific element for the study. At least the changes in body weight should be recovered. The quantity of physical exercise should be reported. As sustained by Russel, RD. (2022 Nutrients) Evidence of nutrition education programs for adults with neurological diseases is lacking.
Author Response
Thank you again for your valuable comments.
